# The Effect of Proprioception Training on Pain Intensity in Thumb Basal Joint Osteoarthritis: A Randomized Controlled Trial

**DOI:** 10.3390/ijerph19063592

**Published:** 2022-03-17

**Authors:** Raquel Cantero-Téllez, David Pérez-Cruzado, Jorge Hugo Villafañe, Santiago García-Orza, Nancy Naughton, Kristin Valdes

**Affiliations:** 1Physical Therapy Section, Ampliación Campus Teatinos, Faculty of Health Sciences, University of Malaga C/Francisco Peñalosa, 29010 Málaga, Spain; cantero@uma.es; 2FE-17, Hand Research Team, IBIMA Área Tecnología en Salud e Innovación, 29010 Malaga, Spain; santiago.orza@gmail.com; 3IRCCS Fondazione Don Carlo Gnocchi, 20148 Milan, Italy; mail@villafane.it; 4Emergency Department, Hospital Comarcal de La Axarquía, 29700 Vélez-Málaga, Spain; 5Department of Occupational Therapy, Hand Surgery Associates, Olyphant, PA 18447, USA; nancynaughton10@gmail.com; 6Occupational Therapy Faculty, Touro University, Henderson, NV 89014, USA; kvaldesotdcht@gmail.com

**Keywords:** proprioception, joint position sense, thumb osteoarthritis, thumb pain, carpometacarpal joint

## Abstract

A randomized controlled trial of forty-five females over 18 years of age with diagnosis of thumb basal osteoarthritis in their dominant hand and with a minimum pain rating of 4/10 on the Visual Analogue Scale (VAS) during activities of daily living (ADLs) were recruited from March to June 2021. The group receiving proprioception training was compared to routine conservative physiotherapy treatment. The main purpose of this clinical trial is to test the effect of proprioception training on pain intensity in subjects with thumb osteoarthritis. Primary outcome was joint position sense (JPS) for the assessment of CMC proprioception and secondary outcomes were Visual Analogue Scale (VAS) and Canadian Occupational Performance Measure (COPM) for the assessment of patient satisfaction and the Quick-DASH which assessed upper limb function. A block randomization was carried out for the control group (*n* = 22) and experimental group (*n* = 23). Participants and evaluator were blinded to the group assignment. Proprioception training produced a statistically significant reduction in pain post intervention, but this reduction was small (d = 0.1) at the 3-month follow-up. JPS accuracy demonstrated statistically significant differences between the groups (*p* = 0.001) post-intervention and at the 3-month follow-up (*p* < 0.003). Statistically significant differences between means were found in both the Quick-Dash and COPM post intervention (both, *p* < 0.001), as well as at the 3-month follow-up (both, *p* < 0.001). There was a significant time factor for the reduction of pain intensity over time but effect sizes between groups was small at the 3-month follow-up period. Proprioceptive training improves thumb JPS accuracy; however, it does not contribute to a reduction in pain intensity in the long term. The inclusion of a proprioceptive program may be beneficial for improving individuals with thumb CMC OA sensorimotor performance. The study was registered at ClinicalTrials.gov NCT04738201. No funding was provided for this study.

## 1. Introduction

Although the pathogenesis of carpometacarpal (CMC) osteoarthritis is not understood completely, altered motor patterns exhibited during the completion of activities of daily living (ADL) can increase symptoms such as thumb pain and decreased pinch strength and thumb mobility [1,2]. Altered motor control patterns that impact one’s activity and participation have been reported in other painful musculoskeletal conditions, including neck pain [3] and knee osteoarthritis [4]. The thumb is one of the most mobile joints of the body and it requires optimal range of motion during ADL performance. There is evidence to support a multi-faceted approach for individuals with thumb CMC osteoarthritis (OA) for reducing pain and improving upper limb function [5,6].

Proprioception and the neuromuscular control of the thumb are essential for maintaining normal joint stabilization to avoid joint deformity or injury [1]. Altered proprioception sense can be related to pain, fatigue, localized nerve or tissue damage, desensitization of the nervous system, changes to the cortical representation of the thumb, or a combination of these factors [7]. These factors can contribute to the degenerative pathology of thumb CMC OA. Therefore, a rehabilitation program that addresses proprioceptive deficits should be considered for individuals with progressive clinical symptoms of thumb CMC OA. The proprioceptive training approach, which includes the subcategories of kinesthesia and joint position sense (JPS), have been included in rehabilitation programs over the last decade [8,9,10]. The role of proprioception training has not been well described in context to the thumb CMC joint; however, the importance of functional thumb motion and pinch strength in everyday activities is clearly defined and understood [11].

The relationship between proprioception and joint disease have been previously described in the literature [12,13,14,15]. Thumb movement provides a variety of sensory feedback that is transmitted to the brain through sensory end organs (joint position sense, movement sense, force sense). Previous studies have concluded that it is necessary to integrate convergent inputs to properly assess body configuration. However, it is unknown how the improvement of each aspect of sensory awareness contributes to the recovery of pain and function in individuals with thumb CMC OA [14]. There are no studies that support the intervention of thumb proprioceptive exercises to reduce pain and improve function.

Decreased proprioceptive function has been previously described when patients with thumb CMC OA were compared to healthy subjects [15,16]. Although the influence of diminished proprioceptive function remains unknown, the presence of a diminished proprioceptive process could be a potential contributing factor in thumb pain [17,18].

To improve the understanding of the influence of conservative interventions, it is necessary to investigate the degree of contribution that each technique, described in the literature, has on the mechanisms that cause pain. The main objective of this investigation is to determine the effect a proprioceptive joint position sense training program has on pain intensity in patients with the diagnosis of thumb CMC joint OA. Secondarily, we will assess the effect of proprioception training on JPS and function.

## 2. Materials and Methods

A randomized controlled single-blind clinical trial study was conducted at a local Hand Rehabilitation Center for patients diagnosed with thumb CMC joint OA referred from three different local hospitals. Informed consent was obtained from all patients and procedures were conducted according to the Declaration of Helsinki and approved by a Local Ethical Committee (62-2020-H).

### 2.1. Participants

Patients with thumb CMC joint OA in their dominant hand were recruited from March to June 2021. Patients who agreed to participate in the study had their thumbs evaluated individually by a rheumatologist and a physiotherapist who used both clinical and radiographic methods. Inclusion criteria included females over 18 years of age, a diagnosis of grade one to three thumb CMC joint OA according to the Eaton Classification [19] in their dominant hand, a reported minimum pain rating of 4/10 on the Visual Analogue Scale (VAS) during activities of daily living (ADLs) at the time of the initial evaluation, and an ability to read and understand the patient information sheets and exercises. Exclusion criteria included subjects who had received treatment for hand or thumb pain in the same limb in the last 6 months, those with a neurological disorder affecting the upper limb, previous surgery to the wrist or hand, fracture, significant hand/thumb injury in the last 6 months, or those who possessed a cognitive impairment that inhibited an understanding of the informed consent or exercise program.

### 2.2. Interventions

Before randomization and after the patient signed the informed consent to participate in the study, a researcher who was blinded to the study collected the baseline outcome measurements. The blinded hand therapist took measurements at baseline, immediately following the 4-week treatment period, and finally at the 3-month follow-up. Data collection, demographic information, and outcome variables were inputted into an Excel database. A number was assigned to each participant to ensure anonymity. Following baseline data collection, the blinded researcher carried out the process of randomization. In order to ensure a balance in sample size across groups over time, a block randomization was carried out for the control group (*n* = 22) and experimental group (*n* = 23).

The treatment was provided by a hand therapist with 20 years of experience as a specialist in the musculoskeletal approach. All participants received a short, hand-based thumb orthosis (Figure 1) to wear at night for the entirety of the study.

Both groups received the identical conservative intervention of a 4-week exercise program. The subjects performed 3 sets of 10 repetitions of pain-free exercises. The intervention consisted of manual distraction of the CMC joint (Figure 2a), relaxation of the adductor thumb muscle by means of a targeted muscle massage (Figure 2b), and active and/or resistive exercises for the first dorsal interosseous (FDI) muscle (Figure 2c) The exercises were carried out in individual treatment sessions 3 times a week. Following each session, patients were encouraged to repeat the exercises one time per day at home.

To improve adherence to treatment and to be able to monitor the adherence of the patient’s performance of the exercise program, an iPad application was used that demonstrated each exercise and recorded the patient’s performance of the exercises. Weekly, the main investigator received an email from the exercise application with a summary of the number of exercise sessions and the number of repetitions performed by the patients.

In addition to the traditional treatment, the experimental group also carried out a proprioceptive exercise program that consisted of reproduction of active joint positions of the thumb. Patients were instructed to reproduce different thumb movements previously taught to by the therapist. For example, move their thumb to the position as had been previously demonstrated (Figure 2d), make movements of the thumb in different degrees and/or stop at the numbers indicated by the therapist on a stick (Figure 2e), and pass a marble along the radial side of the index finger using only the thumb following a specific line (Figure 2f) (in this exercise the first interosseous is also exercised). Figure 2 shows a summary of the exercises.

The testing protocol and assessment protocol was prepared according to the editorial form of medical publishing and CONSORT (Consolidated Standards of Reporting Trials) publishing guidelines [20].

### 2.3. Outcomes

For JPS testing, the participants were seated with the arm resting on a table in a neutral position and elbow at 90 degrees. The joint angle was measured using a standard plastic goniometer. The fulcrum of the goniometer was placed over the intersection of the first and second metacarpals [16,21,22]. The testing was performed with the patient’s eyes closed. The therapist moved the patient’s thumb passively to 30 degrees of palmar abduction. The goniometer was removed, and the patient was asked to hold the position for 3 s. Next, the therapist asked the patient to perform full active thumb adduction and subsequently return the thumb to the initial position (30° CMC abduction). Once participants confirmed the position, a second measurement was taken. The difference between the target angle and the reproduced angle was used to determine the JPS deficit. The greater the angular difference, the greater the JPS deficit. If no differences were obtained between the initial position and the final position, a zero value was assigned. Positive or negative values were assigned for angular differences greater or lesser than the target angle. For data analysis, we used the mean value of two measures. JPS testing has been found to have clinical relevance in its use for individuals with thumb CMC joint OA [16].

The visual analogue scale (VAS) was used to evaluate pain intensity during ADL. The VAS scale has been widely used in diverse adult populations, including those with rheumatic diseases [23]. The sensitivity and reliability of the VAS outcome measure are well defined [24,25] in patients with chronic inflammatory or degenerative joint pain. The VAS has demonstrated sensitivity to changes in pain [26,27,28,29]. All patients were asked to quantify the level of pain experienced with the performance of ADL tasks in the last week using the VAS scale, with 0 indicating the absence of pain and 10 the most extreme pain.

The Quick-DASH questionnaire which is well studied with established validity, reliability, and responsiveness was used to measure upper extremity function [30]. This instrument consists of 11 items providing a total score ranging from 0 to 100 where 0 indicates no limitation and 100 suggests full disability. Occupational performance was measured with the Canadian Occupational Performance Measure (COPM) [31]. The COPM has good convergent validity and responsiveness for evaluating the relationship between patient self-perception and satisfaction for patients with CMC thumb OA [32] and allows subjects to identify goals and engage in a client-specific therapeutic process.

### 2.4. Sample Size Determination

The sample size and power calculations were performed with the ENE 3.0 software (GlaxoSmithKline^©^, Universidad Autónoma, Barcelona, Spain). The calculations are based on detecting a mean difference of 2 cm minimal clinically important difference (MCID) on a 10 cm VAS assuming a standard deviation of 2 cm, a 2-tailed test, an alpha level of 0.05, and a desired power of 80%. The estimated desired sample size is 15 individuals per group. To accommodate for expected dropouts before study completion, a total of 22 and 23 participants for control and experimental group were included.

### 2.5. Statistical Analysis

Data were analyzed using SPSS for Windows (V.25, IBM, Armonk, NY, USA), and an intention-to-treat analysis was conducted using the last-value-forward method. The results are expressed as means, standard deviations, and/or 95% confidence intervals. A separate 2 × 3 mixed model ANOVA, with group (experimental, control) as the between-subjects factor and time (baseline, post-intervention, and 3 months follow-up) as the within-subjects factor, was conducted to examine the effects of the intervention on VAS, JPS, Quick-Dash, and COPM variables. Post-hoc comparisons were conducted with Bonferroni corrections. Between-group effect sizes were calculated by using Cohen’s d coefficient which considers an effect size greater than 0.8 large, 0.5 moderate, and less than 0.2 small [33]. The statistical analysis was conducted at a 95% confidence level and a *p* < 0.05 was considered statistically significant.

## 3. Results

### 3.1. Participants

Sixty-one participants were initially assessed for eligibility. Forty-five consecutive patients who met the inclusion criteria were randomized into either the experimental group (23) or control group (22). At the final data collection time point at 3-months post treatment, 34 patients (mean ± SD age: 62 ± 7 years) were analyzed. The CONSORT flow diagram was followed to structure the progression of patients through the phases of this study (Figure 3). Baseline characteristics of the patients in each group are presented in Table 1 and the first two columns of Table 2. The Kolmogorov–Smirnov test showed a normal distribution of the data.

### 3.2. Paint Intensity

There was a significant time factor (F = 224.073; *p* < 0.001) for the reduction of pain intensity as measured by the VAS over time. Post-hoc analysis indicated that patients with thumb CMC OA receiving the experimental treatment had a statistically significant reduction in pain in post-intervention (experimental group mean, 5.7; 95% CI: 5.4; 6.1, control group mean, 6.4; 95% CI: 6.1; 6.8, *p* < 0.001; significant difference between groups 0.7; 95% CI: −1.2; 0.2, *p* = 0.003) period, as well as at 3-month follow-up (experimental group mean, 5.2; 95% CI: 4.9; 5.5, control group mean, 5.3; 95% CI: 4.9; 5.6, *p* = 0.005; there was no significant difference between the effect size of the interventions between groups, 0.1; 95% CI: 1.7–2.6, *p* = 0.8) (Table 2). Between-groups effect sizes were large (d = 1.0) after the intervention and small (d = 0.1) at 3-month follow-up period.

### 3.3. Joint Position Sense

Outcome for JPS demonstrated a significant time factor (F = 53.071; *p* < 0.001) over the length of the study. We found statistically significant differences in the experimental group (*p* = 0.001) post-intervention and at 3-months follow-up (*p* < 0.003). There was significant difference between both groups at 3-months follow-up (*p* < 0.001). Between-groups effect sizes were moderate (d = 0.5) after the intervention and large (d = 1.8) at the 3-month follow-up period (Table 2).

### 3.4. Function and Satisfaction

Outcomes for the Quick-Dash and COPM demonstrated a statistically significant difference between means over time (F = 77.856 to 496.808, both *p* < 0.001). The post-hoc testing revealed significant decreases in the scores of the Quick-Dash and COPM in both groups at post-intervention when compared to baseline data (both, *p* < 0.001), as well as at 3-months follow-up (both, *p* < 0.001). There was also no significant main effect difference between the groups at post-intervention and at 3-months follow-up (*p* > 0.05) (Table 2).

## 4. Discussion

This is the first clinical trial that reports findings of the effects of proprioceptive joint sense training on outcomes for patients with a diagnosis of thumb CMC joint OA. There is no high-level evidence defining the best techniques or exercises for proprioceptive rehabilitation for thumb CMC joint OA. The thumb exercise program used in this experiment was based on a previous study [34].

Pain is often the primary reason patients seek rehabilitative services and can be the most influential factor on the performance of daily tasks and functional activities. Therefore, investigating the effect that conservative techniques have on pain intensity is warranted. Manual therapy, orthotic intervention, and a multi-model approach have evidence to support their use to improve function and decrease pain [35,36]. We found a reduction in pain with a clinically significant difference at the 3-months follow-up of greater than a two-point reduction in the VAS [37]. The decrease in pain intensity occurs in both groups but the effect size between groups is not significant and is further reduced at the 3-month follow-up. Therefore, we can speculate that the effect in the initial phase may be associated with the thumb orthosis as opposed to the proprioception exercises.

Thumb OA is a progressive pathology that produces chronic pain and affects all areas of one’s life. Some studies on thumb CMC OA have used intervention techniques that are not based on the chronic pain approach [38,39]. OA involves complex mechanisms of altered pain transmission and is related to innumerable internal and external factors [40].

Some studies that investigated chronic pain concluded that people with chronic pain have a proprioceptive deficit [40]. A study by Seok and colleagues [15] revealed that patients diagnosed with thumb OA have decreased proprioceptive function which is associated with the presence of osteoarthritis. The effect of proprioceptive training on pain intensity was not maintained over time in our study or in previous studies that performed similar conservative interventions of orthotic intervention, exercise, or ADL re-education [41,42,43]. Based on other research, proprioceptive work may help to recover the osteoarticular balance and coordination and improve the performance of daily activities and consequently decrease pain intensity [44,45,46].

It is important to consider intervention strategies that focus on the central origin of pain; education on ADLs, physical and psychological approaches, and force sense proprioception have not yet been studied for efficacy in thumb CMC OA. To prevent interference with the result of the joint proprioceptive intervention, in our clinical trial, we did not include ADL re-education although some studies recommend re-education in lifestyle performance in individuals with hip and knee osteoarthritis [47]. Without adequate information and advice from healthcare professionals, people with thumb OA may not know how to perform their ADL tasks without pain and may avoid activities so as not to provoke pain. Nevertheless, patient education programs on adaptive ADL techniques should be included in the rehabilitation program of individuals with CMC OA.

We did not find significant differences between groups for pain or function. We did find statistically significant differences regarding JPS testing between groups following the intervention and at the 3-month follow-up. The MCID for JPS for CMC OA has not been established. These results confirm those obtained in the pilot study carried out by Cantero-Tellez et al. [34]. However, the improvement in JPS testing may be due in part to the learning that occurred with repetition of the positioning exercises during the intervention, rather than an improvement in JPS.

Exercises and education regarding pain have been found to be beneficial for pain intensity, self-efficacy, and social function [48]. Despite having obtained positive results in terms of patient satisfaction and pain, our intervention program that included proprioceptive exercises did not allow us to reach solid conclusions regarding the effectiveness of JPS exercises on pain intensity or patient satisfaction in the long term. We have not taken into consideration other aspects that could influence the perception of pain, such as anxiety or depression [49,50]. Biopsychosocial factors are associated with patient-reported upper extremity disability, and there is a focus on how depression and/or anxiety are associated with functional improvements in patients that have chronic musculoskeletal pain [50]. We have not determined how anxiety and depression may impact the functional performance of individuals with thumb CMC OA because we did not assess depression or anxiety in our study. 

There is evidence to support JPS as a clinically meaningful measure of conscious sensorimotor impairment in individuals with thumb CMC OA [17,35] but according to our results, we cannot establish a relationship between the improvement in JPS and the intensity of pain in patients with thumb CMC OA. A longer follow-up period of 6–12 months could help determine the long-term efficacy of the intervention. Future research studies should consider performing a sole intervention of JPS training or force sense proprioception exercises to determine outcomes over time in the thumb CMC OA population. Additionally, studies with more reliable objective measurements regarding JPS such as electrogoniometers or wearable sensors would be beneficial. It would also be worthwhile to study the effectiveness of different proprioceptive interventions to determine the most appropriate proprioception exercises for thumb osteoarthritis. It would also be advantageous to determine the effect of motor re-education on the performance of ADL tasks and pain intensity using a multi-modal pain treatment approach.

### Study Limitations

Our follow-up time was 12 weeks. It would be beneficial to determine the efficacy of the intervention at a longer follow-up period of 6 months to 12 months. We also provided a multi-modal intervention, and it is difficult to determine the exact impact that proprioception training had on the outcomes. 

## 5. Conclusions

Patients with a diagnosis of thumb CMC OA made clinically significant improvements over time in pain, function, and satisfaction in both groups, but the effects on pain intensity decreased at the 3-month follow-up. The group receiving proprioception exercises achieved both clinically and statistically significant changes in JPS error scores, suggesting that the inclusion of a proprioceptive program may be beneficial for improving sensorimotor performance.

## Figures and Tables

**Figure 1 ijerph-19-03592-f001:**
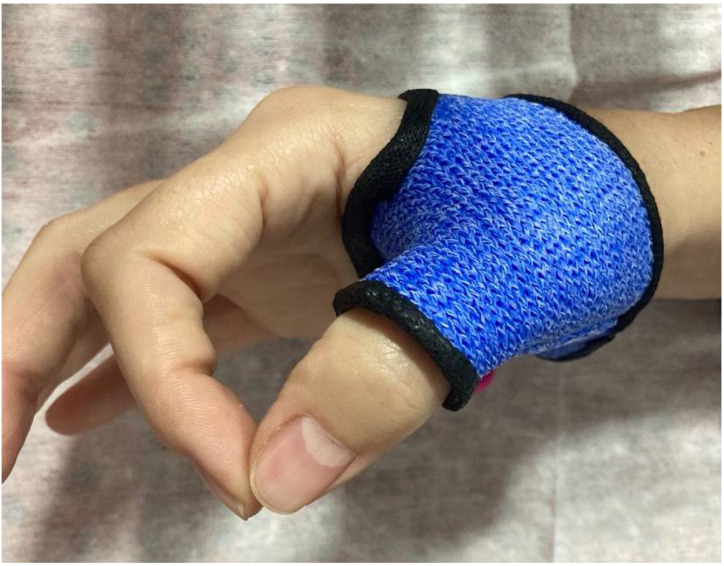
Short thumb orthosis.

**Figure 2 ijerph-19-03592-f002:**
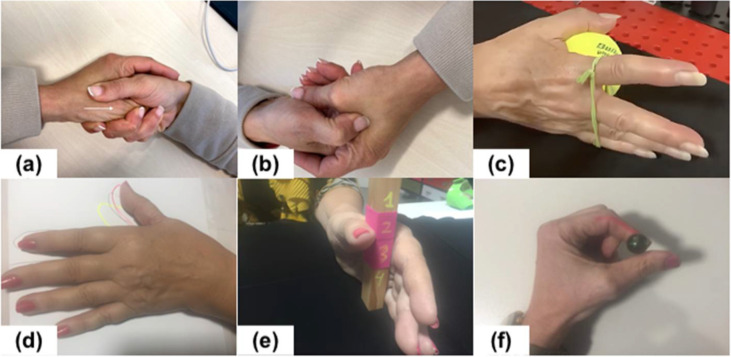
Movement to be reproduced by patients. (**a**) manual distraction of the CMC joint; (**b**) massage of the adductor thumb muscle; (**c**) resistive exercises for the first dorsal interosseous; (**d**) move their thumb to the position as had been previously demonstrated; (**e**) movements of the thumb in different degrees and/or stop at the numbers indicated by the therapist on a stick; (**f**) pass a marble along the radial side of the index finger.

**Figure 3 ijerph-19-03592-f003:**
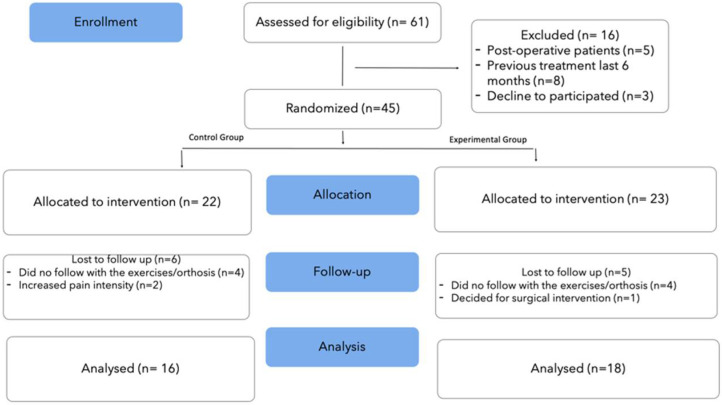
Flow chart of the study.

**Table 1 ijerph-19-03592-t001:** Characteristics of patients at baseline.

Characteristics	Exp (*n* = 18)	Con (*n* = 16)
Age (years), mean (SD)	63 (7)	62 (7)
Dominant hand, right %	18 (86%)	18 (95%)
Affected hand, right %	17 (81%)	17 (90%)
Outcomes		
VAS, mean (SD)	7.6 (1.0)	7.1 (0.9)
JPS, mean (SD)	9.1 (2.9)	9.0 (3.6)
*Quick*DASH, mean (SD)	64.7 (6.7)	63.9 (7.1)
COMP, mean (SD)	2.3 (0.8)	2.6 (0.7)

Exp = experimental group, Con = control group, VAS: visual analog scale, JPS = Joint position sense, COMP = Canadian Occupational Performance Measure, Dash: Disabilities of the Arm, Shoulder and Hand Scale.

**Table 2 ijerph-19-03592-t002:** Pain intensity, joint position sense, and quality of Life. Mean (SD) for outcomes at all study visits for each group, mean (SD) difference within groups, and mean (95% CI) difference between groups.

Outcome	Group						Difference within Groups				Difference between Groups	
	Pre		Post		FU		Post Minus Pre		FU Minus Pre		Post	FU
	Exp	Con	Exp	Con	Exp	Con	Exp	Con	Exp	Con	Exp Minus Con	Exp Minus Con
	(*n* = 18)	(*n* = 16)	(*n* = 18)	(*n* = 16)	(*n* = 18)	(*n* = 16)	(*n* = 18)	(*n* = 16)	(*n* = 18)	(*n* = 16)	(*n* = 34)	(*n* = 34)
VAS	7.4 (1.0)	7.4 (0.8)	5.7 (0.7)	6.4 (0.9)	5.2 (0.5)	5.3 (0.7)	−1.7 * (0.2)	−1.0 * (0.2)	−2.2 * (0.2)	−2.1 * (0.2)	−0.7 # (−1.2 to −0.2)	−0.1 (−0.5 to 0.4)
JPS	10.6 (4.5)	10.6 (4.8)	6.8 (4.2)	8.8 (3.9)	2.7 (2.4)	7.5 (2.6)	−3.7 * (0.6)	−1.9 * (0.6)	−7.9 * (0.7)	−3.1 * (0.8)	−1.9 (−4.8 to 0.9)	−6.6 # (−6.6 to −3.1)
*Quick*DASH	65.3 (5.6)	64.4 (6.4)	60.8 (5.9)	62.1 (6.6)	59.0 (5.4)	59.8 (76.6)	−4.4 * (0.7)	−2.3 * (0.7)	−6.3 * (0.7)	−4.6 * (0.7)	−1.3 (−5.6 to 3.1)	−0.8 (−5.0 to 3.4)
COPM	2.3 (0.7)	2.4 (0.6)	5.0 (0.7)	3.7 (0.7)	6.6 (0.6)	5.5 (0.3)	2.7 * (0.2)	1.3 * (0.2)	4.3 * (0.2)	3.1 * (0.2)	1.3 (0.9 to 1.8)	1.1 (0.8 to 1.5)

Exp = experimental group, Con = control group, FU = Follow-up, VAS: visual analog scale, JPS = Joint position sense, COMP = Canadian Occupational Performance Measure, DASH: Disabilities of the Arm, Shoulder and Hand Scale * Significant difference within-group, # Significant difference between-group, *p* < 0.05 (95% confidence interval).

## Data Availability

Deidentified data are available upon written request to the corresponding author.

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
