# Peer review of "The Effect of Proprioception Training on Pain Intensity in Thumb Basal Joint Osteoarthritis: A Randomized Controlled Trial"

_ijerph, 2022, doi:10.3390/ijerph19063592_

Round 1

Reviewer 1 Report

The study is well conducted.

It is requested to improve the image quality of the flowchart. It cannot be displayed correctly in the paper. Also the quality resolution of Figure 1 and 2 could be increased.

The duration of each individual treatment is not specified in the text. The rehabilitation protocol represents the main body of this study so it may be useful to insert a summary table of the passive, active-assisted and active mobilization exercises that have been included in the sessions and the differences between the two groups.

Was there a difference in the duration of treatment of each single session of the two groups? If so, how long was it?

It may be interesting to use more reliable objective measurements regarding JPS in the future such as electrogoniometers or wearable sensors.

The implementation of the study of grip strength can be useful in defining an optimal rehabilitation program in CMC osteoarthritis.

At line 90 please insert effect "of" a proprioceptive joint...

Best Regards

Author Response

RESPONSE TO REVIEWERS

REVIEWER 1. The study is well conducted.

It is requested to improve the image quality of the flowchart. It cannot be displayed correctly in the paper.

Also the quality resolution of Figure 1 and 2 could be increased.

Thank you, we have improved resolution of the Flowchart and figures

The duration of each individual treatment is not specified in the text.

Is specified in line 143 when we said: “Both groups received the identical conservative

intervention of a 4-week exercise program”

The rehabilitation protocol represents the main body of this study so it may be useful to insert a summary table of the passive, active-assisted and active mobilization exercises that have been included in the sessions and the differences between the two groups.

A summary of the control group and proprioception program is included in a new fig 2 in order to clarify the exercises follow up for each group. We have also clarified it in the methodology section.

Was there a difference in the duration of treatment of each single session of the two groups? If so, how long was it?

Each single session had a duration more or less of 15-20 min for control group and 20-30 min for Experimental group.

It may be interesting to use more reliable objective measurements regarding JPS in the future such as electrogoniometers or wearable sensors.

Of course, you are right, we have added your suggestion on line 391 as follow: Future research studies should consider performing a sole intervention of JPS training or force sense proprioception exercises to determine outcomes over time in the thumb CMC OA population and study more reliable objective measurements regarding JPS such as electrogoniometers or wearable sensors”

The implementation of the study of grip strength can be useful in defining an optimal rehabilitation program in CMC osteoarthritis.

Yes, you are right. It would be also interesting to investigate the effect of the key or point-to-point pinch. However, the ability to carry out grip or pinch force, pain intensity and upper limb function are closely correlated. We are currently carrying out specific studies to verify the effect of proprioceptive work and the perception of the patient's strength. Thank you for your very interesting appreciation.

At line 90 please insert effect "of" a proprioceptive joint...

Done

Reviewer 2 Report

The role of proprioception training in thumb basal joint osteoarthritis was studied in the manuscript. The topic is potentially interesting, although significant difference at the 3-month follow-up was not found by the proprioception training. The discussion should be strengthened why the pain improved in the short-term, but not in the long-term.

Line 34-36: Is this sentence correct? Significant difference between groups in VAS is shown post intervention in Table 2.

Line 133: Does “abduction” mean “radial abduction” or “palmar abduction”?

Line 134: Does “for 3 s” mean “for 3 seconds”?

Line 187-196: The description of the proprioceptive exercise is not specific enough. Please explain the details of the movements shown in Figure 2.

Discussion:  Is it possible to interpret the results that proprioceptive exercise may speed up the improvement of pain? If so, please speculate why?

Line 389-393: Two references are shown as reference 1. One needs to be deleted.

Author Response

REVIWER 2.

The role of proprioception training in thumb basal joint osteoarthritis was studied in the manuscript. The topic is potentially interesting, although significant difference at the 3-month follow-up was not found by the proprioception training. The discussion should be strengthened why the pain improved in the short-term, but not in the long-term.

Thank you for your advice. Given that the follow-up time of the patients has been three months, it does not allow us to know exactly if the effects on the intensity of pain are maintained in the long term or even if they are maintained once the patient no longer continues with the exercises. It is something that we will have to value in the future. However, in the paragraph between lines 369-383 we explain other considerations that could make the long-term effects not good, we have added the phrase:”….. positive results in terms of patient satisfaction and pain, do not allow us to reach solid conclusions regarding the effectiveness of JPS exercises on pain intensity or patient satisfaction in the long-term. We have not taken into consideration other aspects that could influence the perception of pain, such as anxiety or depression….”

Line 34-36: Is this sentence correct? Significant difference between groups in VAS is shown post intervention in Table 2.

We have clarified and change as follow: “There was a significant time factor for the reduction of pain intensity over time but effect sizes between groups was small at the 3 month follow-up period”

Line 133: Does “abduction” mean “radial abduction” or “palmar abduction”?

Palmar abduction. We clarified in text.

Line 134: Does “for 3 s” mean “for 3 seconds”?

Yes we have change 3s for 3 seconds

Line 187-196: The description of the proprioceptive exercise is not specific enough. Please explain the details of the movements shown in Figure 2.

Ok, we have added a paragraph to clarified in line 225 as follow: “In addition to the traditional treatment, the experimental group also carried out a proprioceptive exercise program that consisted of reproduction of active joint positions of the thumb. As show in figure 2, patients were instructed to reproduce different thumb movements previously taught to by the therapist. atients were instructed to reproduce different thumb movements previously taught  by the therapist. For example, pass a marble along the radial side of the index finger using only the thumb following a specific line (in this exercise the first interosseous is also exercised), make movements of the thumb in different degrees and/or stop at the numbers indicated by the therapist on a stick among others as shown in Figure 2. Next the patient was asked to actively move their thumb to the position as had been  previously demonstrated and performed in their therapy sessions. Their thumb was then  returned to the initial start position actively. Patients  were required to reproduce the target thumb position previously experienced but with their eyes closed. Table 1 shows a summary of the exercises for each group.”

Discussion:  Is it possible to interpret the results that proprioceptive exercise may speed up the improvement of pain? If so, please speculate why?

Is a good question… I don’t think so, as we say in the discussion:”…. we can speculate that the effect in the initial phase may be associated with the thumb orthosis as opposed to the proprioception exercises”. Instead, proprioceptive work may help to recover the osteoarticular balance and coordination and improve the performance of the daily activities as have already describe in previous studies in OA, in consequently, decrease pain intensity, but we can not concluded that a decrease on pain intensity is because proprioceptive exercises, future investigation should determinate it. We did find statistically significant differences regarding JPS testing between groups following the intervention and at the 3-month follow-up, but we did not find significant differences between groups for pain or function.

Line 389-393: Two references are shown as reference 1. One needs to be deleted.

Done

Reviewer 3 Report

Suggestions for authors:

Please follow http://www.consort-statement.org/media/default/downloads/CONSORT%202010%20Checklist.pdf and http://www.congrex.ch/fileadmin/files/2012/eccmid2012/ECCMID12_CONSORT_abstract_%20checklist.pdf

Please add: Participants: Eligibility criteria for participants and the settings where the data were collected; Interventions intended for each group. Outcome Clearly defined primary outcome for this report; Randomisation and How participants were allocated to interventions.. Blinding (masking) Whether or not participants, care givers, and these assessing the outcomes were blinded to group assignment.

The methods are confusing and not divided correctly - follow the checklist and help the reading
Methods
Trial design Description of trial design (such as parallel, factorial) including allocation ratio

Participants - Eligibility criteria for participants; Settings and locations where the data were collected
Interventions The interventions for each group with sufficient details to allow replication, including how and when they were actually administered (a figure, I think is necessary?). This section is the key of the manuscript, and I recommend a major revision
Outcomes - Completely defined pre-specified primary and secondary outcome measures, including how and when they were assessed

Sample size ? How sample size was determined …When applicable, explanation of any interim analyses and stopping guidelines

Figures 1 and 2 are really of poor quality and for this reason they almost reduce the value of the intervention. I highly recommend using a figure to explain the two different interventions ... maybe you describe the orthosis with a drawing .. maybe even for figure two , with an illustration the methodological rationale could be clearer.

In results:

I would suggest a baseline t test for comparing the two groups at T0.

Harms, All important harms or unintended effects in each group are missing

Author Response

REVIWER 3.

 Please follow http://www.consort-statement.org/media/default/downloads/CONSORT%202010%20Checklist.pdf and http://www.congrex.ch/fileadmin/files/2012/eccmid2012/ECCMID12_CONSORT_abstract_%20checklist.pdf

Thank you for your advised and correction. Done.

Please add: Participants: Eligibility criteria for participants and the settings where the data were collected; Interventions intended for each group. Outcome Clearly defined primary outcome for this report; Randomisation and How participants were allocated to interventions.. Blinding (masking) Whether or not participants, care givers, and these assessing the outcomes were blinded to group assignment.

Done

The methods are confusing and not divided correctly - follow the checklist and help the reading

Methods 
Trial design Description of trial design (such as parallel, factorial) including allocation ratio

Participants - Eligibility criteria for participants; Settings and locations where the data were collected 
Interventions The interventions for each group with sufficient details to allow replication, including how and when they were actually administered (a figure, I think is necessary?). This section is the key of the manuscript, and I recommend a major revision

Thank you very much for your advice that helped improve the paper. We have made sub-divisions in the methodology following your instructions and we have specified in more detail the exercises carried out by the participants, accompanying it with a new table 1 where you can see the exercises carried out by the control and experimental groups.

Outcomes - Completely defined pre-specified primary and secondary outcome measures, including how and when they were assessed

We specified in the first paragraph of the intervention (2.2), “baseline measurements, and immediately after the 4-week treatment period, and finally at the 3-month follow up”

In the abstract the primary and secondaries outcomes are specified, in that same order each one of them is explained in the methodology, specifying in each of them how it comes measures.

Sample size ? How sample size was determined …When applicable, explanation of any interim analyses and stopping guidelines.

:We’ve have completed as follow: “The sample size and power calculations were performed with the ENE 3.0 software (GlaxoSmithKline©, Universidad Autónoma, Barcelona). The calculations are based on detecting a mean difference of 2cm minimal clinically important difference (MCID) on a 10cm VAS assuming a standard deviation of 2cm, a 2-tailed test, an alpha level of 0.05, and a desired power of 80%. The estimated desired sample size is 15 individuals per group. To accommodate for expected dropouts before study completion, a total of 22 and 23 participants for control and experimental group were included ”

Figures 1 and 2 are really of poor quality and for this reason they almost reduce the value of the intervention. I highly recommend using a figure to explain the two different interventions ... maybe you describe the orthosis with a drawing .. maybe even for figure two , with an illustration the methodological rationale could be clearer.

Ok, done, we have add figures for both interventions.

In results:

I would suggest a baseline t test for comparing the two groups at T0.

Harms, All important harms or unintended effects in each group are missing.

We’ve have done, thanks (table 2 and the first two columns of table 3)

Round 2

Reviewer 2 Report

The manuscript is now revised accordingly.

Line 160: Is "atients" "Patients"?

Line 423-427; Ref 1 is not corrected. There are still 2 articles as ref 1 ("Necessity of....." and "Development of the QuickDASH.....").

Author Response

Thank you very much for helping us to improve the manuscript.

Line 160 is Patients. Change

The second reference corresponding to the reference 29 has been deleted from reference 1.

Reviewer 3 Report

Implementation of the text makes the work appropriate for publication. However, the need to include better-quality photographic images is reiterated.

Author Response

Thank you very much for helping us to improve the manuscript. The images were uploaded to the platform and sent to the editor, however they have not been incorporated into the manuscript. I enclose the manuscript with the new images inserted, both the images of the exercises and the flowchart.
